# The role of the working memory storage component in a random-like series generation

**Mikołaj Biesaga**[1]*, **Andrzej Nowak**[1,2]

**1** Robert Zajonc Institute for Social Studies, University of Warsaw, Warsaw, Poland, **2** Department of Psychology, Florida Atlantic University, Boca Raton, Florida, United States of America

* m.biesaga@uw.edu.pl

**Data Availability Statement:** Data and scripts for statistical analysis are available on Open Science Foundation at https://www.doi.org/10.17605/OSF.IO/CK78N.

## Abstract

People are not equipped with an internal random series generator. When asked to produce a random series they simply try to reproduce an output of known random process. However, this endeavor is very often limited by their working memory capacity. Here, we investigate the model of random-like series generation that accounts for the involvement of storage and processing components of working memory. In two studies, we used a modern, robust measure of randomness to assess human-generated series. In Study 1, in the experimental design with the visibility of the last generated elements as a between-subjects variable, we tested whether decreasing cognitive load on working memory would mitigate the decay in the level of randomness of the generated series. Moreover, we investigated the relationship between randomness judgment and algorithmic complexity of human-generated series. Results showed that when people did not have to solely rely on their working memory storage component to maintain active past choices they were able to prolongate their high-quality performance. Moreover, people who were able to better distinguish more complex patterns at the same time generated more random series. In Study 2, in the correlational design, we examined the relationship between working memory capacity and the ability to produce random-like series. Results revealed that individuals with longer working memory capacity also were to produce more complex series. These findings highlight the importance of working memory in generating random-like series and provide insights into the underlying mechanisms of this cognitive process.

## Introduction

Very often people see patterns in events that occurred by chance even if the process that was responsible for generating them was indistinguishable from truly random one. This occurs whether the events are life-threatening, such as the location of the next bomb strike [1], or relate to more trivial matters, such as the order of songs in a music device [2], sports events like shooting a basket [3], or outcomes of a roulette wheel [4]. Regardless of the domain, individuals tend to expect that random series will have more alternations than repetitions of consecutive elements. This leads them to believe that lightning will not strike twice in the same place

**Funding:** This work was supported by funds from the Polish National Science Centre (project no. 2019/35/N/HS6/04318). The funders had no role in study design, data collection, and analysis, decision to publish, or preparation of the manuscript.

**Competing interests:** The authors have declared that no competing interests exist.

or that in a fair coin toss, the probability of flipping tails will increase as the run of heads gets longer [5]. These common intuitions may reflect a tendency to impose order on randomness, despite the lack of evidence for a non-random pattern.

In the literature on randomness judgments [6, 7], this phenomenon is often referred to as either gambler's fallacy or negative recency—a tendency to overestimate the frequency of alternations. For example, in experiments in which the task was to assess the outcomes of consequent fair coin tosses, people considered series with alternation rates between 0.6 and 0.7 more random than series with an exact alternation rate of 0.5 [8–10]. Moreover, when producing a random series, people not only violate the constraint of the uniform frequencies of distinct elements but also their series exhibit interdependence of consequent items. Therefore, they tend to underestimate the frequency of runs [11–13]. Consequently, in human-generated data, the probability of repeating the same element is lower than it should be if each item were treated as an outcome of an independent event. In other words, when asked to produce a random series of fair coin tosses people perceive it as more likely to change from heads to tails (or the other way around) than to repeat the previous item.

These biases observed in human-generated data traditionally were either attributed to the task itself or the intrinsic constraints [14, 15]. Ayton et al. [16] showed that in many classic studies on random-like series generation, the instruction promoted a certain type of answers that prevented participants from creating independent responses. Moreover, Kareev [17] argued that regardless of the task/instruction people tend to generate sequences that they consider the most representative of a random process, rather than the most random ones. Consequently, researchers not only struggled with the task definition but also with the measurement of the randomness for relatively short sequences. Therefore, it was very difficult to compare the results in-between studies [18].

Only recently, Gauvrit et al. [19] proposed algorithmic complexity as the measure of randomness that is applicable to human-generated data and allows to overcome both definitional and methodological problems from which some of the previous studies suffered. It is grounded in algorithmic information theory (AIT) with the notion of Kolmogorov-Chaitin (algorithmic) complexity [20]. It defines a random string as a string that cannot be compressed with any algorithm significantly simpler than the string itself. Such a definition has an intuitive psychological interpretation. That is, to remember a random series one has to remember each element separately because it is difficult to chunk it. For example, the following series 010101 is relatively easy to memorize as an alternation of zeros and ones and it has a normalized algorithmic complexity of approximately .542 (normalized algorithmic complexity is bounded between 0 and 1, where 0 denotes a very simple string of all zeros or ones and 1 a very complex one). On the other hand, the following series with the same proportion of elements and lower alternation rate is more difficult to memorize 101001 and therefore has higher algorithmic complexity – .647. A simple psychological interpretation would be that one has to allocate more cognitive resources to memorize the second series than the first one.

The second group of research attributed the biases in human-generated data to intrinsic constraints. On one hand, the focus was on cognitive constraints, i.e., the limitation of the central executive of working memory in the process of response selection [21, 22], retrieval of random patterns from memory [23], or on the objective difficulties of randomness judgment [24]. Kareev [25] argued that when people generate random-like series they never consider each element separately but instead they maintain active in the working memory only subsequence of recent choices. Therefore, the biases observed in human-generated data might be attributed to the limitation of the working memory storage component. Moreover, Warren et al. [26] demonstrated that when the capacity of working memory is taken into account human-generated series resembled sequences produced by a random process. However, in their analysis, they

only focused on the frequencies of elements in the whole series not investigating how this resemblance changes over time.

On the other hand, research on the organized correlated nature of human behavior showed that human responses in a vast array of domains are structured in non-trivial and non-pattern ways [27]. From the coordination dynamics perspective, the deviations from randomness in human-generated sequences could be explained in terms of bimanual coordination attractors [28]. In two-button random production tasks, the generated sequence might be considered as a result of coordination between two coupled systems [29]. Therefore, altering and repeating sequences might be seen as anti-phase and in-phase attractors. From this perspective, the alternation bias might be seen as the emergence of the relatively stable over-time anti-phase attractor which as in most movement tasks destabilizes with the increased pace of generation [30].

In this article, we aim to focus on the dynamic aspect of random-like series production and analyze the role of working memory in the process of series generation. We use a modern, objective method grounded in AIT to quantify the level of randomness of generated sequences. Compared to methods based on the probability theory, this approach not only provides more accurate estimates but also enables us to track the dynamics of randomness during the task.

The article is organized as follows. In the next subsection, we describe a simple theoretical model of random-like series generation proposed by Biesaga et al. [18] that accounts for the involvement of two components of working memory. Next, in the following sections, based on the existing literature we formulate three hypotheses regarding the role of working memory in the random-like series generation process. Then, in two studies using modern methods for quantifying randomness grounded in AIT and based on the notion of algorithmic (Kolmogorov) complexity, we test the hypothesis regarding the role of the working memory storage component (H1), the effects of individual differences in the randomness judgments (H2) and working memory capacity (H3), on the complexity of human-generated series.

## Random generation model

In general, it is unlikely that people have an innate and universal mechanism to generate truly random series. The vast body of the literature demonstrated that the context in which people are asked to produce a random series (instruction or a task itself) significantly affects their performance [13, 18]. Under the right circumstances, they are even able to generate series that outplay computers in zero-sum games in which the random strategy is the most successful [31, 32]. That is likely because people attempt to simulate the random process accordingly to the context, for example, tossing a fair coin or rolling a die. These simulations are based on their knowledge and personal experience with a given process. Consequently, their approximations of a random process depend on an interaction between exogenous influences and endogenous cognitive control operations. Biesaga et al. [18] argued that the the production of non-trivial, random-like sequences requires an optimal systemic interaction between contextual cues and cognitive constraints. Moreover, this optimal level is highly unique as it reflects individual preferences and inhabitation processes [33].

The model proposed by Biesaga et al. [18] accounts for both bottom-up and top-down processes. It assumes that contextual cues help activate internalized schema of random process outputs that people try to reproduce, while working memory constraints limit the accuracy of the reproduction of a given pattern (see Fig 1). When generating a series, individuals first establish which schema should be activated based on available contextual cues, such as tossing a fair coin. Second, they propose a new element that aligns with past choices and reproduces a known random pattern. This new element is selected from available options based on the active schema. Third, they evaluate whether the addition of the element would increase the

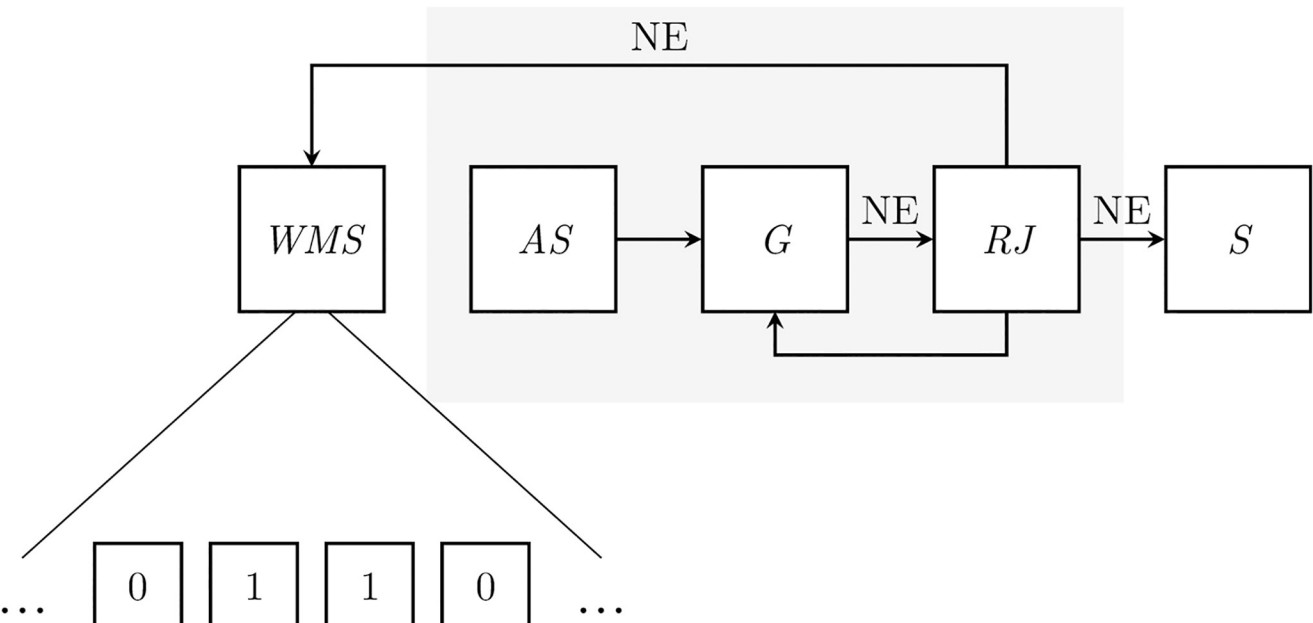

**Fig 1. Schematic model of random series generation.** WMS—Working Memory Storage component, AS—Active Schema, G—Random Series Generator, RJ—Randomness Judgment, S—Output sequence, NE—a new element of the series, elements in the gray rectangle are executed within Working Memory Processing component. During the random series generation process, people maintain only the last few elements, that have been produced, active in their working memory. When a new element is generated, first the schema of random outputs is activated. Second, the new element is proposed. Third, the subsequence stored in the working memory is compared to the random sequence's prototype and based on the result a next element is either selected or discarded (then another one is generated). The new element is appended to the sequence in the working memory, while the first one is removed.

randomness of the series. If not, they propose a new element. This process continues until the addition of the new element to the past choices is perceived as increasing the randomness. Finally, the already generated series is updated with the new element.

In this article, our objective is to empirically investigate the theoretical assumptions of the described above model regarding the cognitive constraints involved in the production of random-like series. Specifically, we seek to examine the influence of both the storage and processing components of working memory on the generation of such sequences, as well as the relationship between working memory capacity and the level of randomness exhibited in human-generated data.

## The role of storage component

In accordance with the Baddeley multicomponent working memory model [34, 35], the model proposed by Biesaga et al. [18] posits that generating random series involves both storing and processing components of the working memory. The storing component is responsible for maintaining active the subsequence of the already generated sequence, while the processing component is employed to evaluate whether the proposed new element increases the randomness of past choices (compare Fig 1). This distinction builds on classic results that demonstrated the involvement of the central executive, phonological loop, and visuospatial activity systems of the working memory during random-like series generation [21]. Biesaga et al. [18] similarly to Vandierendonck et al. [12] argue that the storage component of the working memory must play a distinct role from the central executive system. It allows for maintaining active a subsequence that contains the last few elements of the generated series. However, despite

their distinct roles, both components of the working memory rely on a limited-capacity domain-general central attentional controller. Consequently, because the random-like series generation process requires multiple executive functions for monitoring and updating the working memory storage component, active schema shifting, and set-shifting, it is a cognitively expensive process [36]. As a result, the increased cognitive load on one component affects the performance of the second [37].

Here, we aim to experimentally test the hypothesis that reducing the load on the storage component of working memory would improve the performance of the processing component in evaluating the randomness of generated series. We predict that participants who do not have to maintain active past choice will be able to sustain the initial level of randomness for a longer period of time, compared to those who have to rely on working memory. However, we do not expect this to result in an overall increase in the randomness of the series, as this would require either a change in context or a better schema that people try to reproduce when generating random series [12, 18]. Instead, we anticipate a slower decrease of the randomness level. We still expect the decrease in the performance because of the fatigue due to mundane receptiveness of the task [38].

## The role of processing component

The process of monitoring past choices, proposing a new element accordingly to the activated schema, and testing whether the proposed element increases the randomness of the already generated subsequence assumes the involvement of the processing component of the working memory. Previous research, indeed, demonstrated that the executive control must be employed in the random-like series generation [39]. Moreover, the increased load on the central executive system lowers the level of randomness in produced series [21] For example, Vandierendonck [40] demonstrated that the level of randomness in dual-task random-like series generation deviated from that of single-task production. These findings suggest that people produce less random series when the central executive system is simultaneously engaged in another task, providing further evidence for its involvement in random-like series generation.

Building on prior research, Biesaga et al. [18] proposed a model that posits the involvement of the central executive system of working memory. This system is responsible for monitoring past choices, proposing a new element based on the activated schema, and assessing whether the proposed element enhances the randomness of the already generated series. Specifically, they suggest that when generating a random series, individuals simply decide on the last element and select it from available options in accordance with the active schema. This schema represents the pattern of events that individuals have internalized through experience as being random. This assumption builds on the representativeness hypothesis that stated that when people produce a random series they do not consider all possibilities but only the ones that they perceive the most representative for a random process [12, 41]. Consequently, some of the patterns are underrepresented and therefore overlooked in the generation process. Research on human perception of randomness suggests that people judge the randomness of a series by evaluating its complexity [8], rather than examining its statistical features. Consequently, they tend to avoid producing simple patterns that they associate with non-randomness, leading to an underrepresentation of certain series types in human-generated data. For example, in a task that asks participants to simulate a fair coin toss, the series HTHTHTH is often avoided. Moreover, Falk and Konold [8] also found that human-generated sequences are often indistinguishable from what is considered random. This indicates that the judgment of randomness is a key component in random-like sequence generation.

According to the Biesaga et al [18] model, if the proposed element fails to increase the complexity of the series, individuals select a different element to append to the past choices. Therefore, individuals must possess the ability to discern which subsequences, differing only in their last element, are more random in order to generate random series. This suggests that the ability to distinguish series of higher complexity is a key factor in generating random-like series. Thus, individuals with better ability to distinguish between series of varying complexity are expected to produce more random ones.

## The working memory capacity

The model proposed by Biesaga et al. [18] builds on the assumption made by Rapoport and Budescu [41] that the storage component of working memory in the process of random-like series generation maintains active a limited number of past choices. Based on these active elements processing component makes decision of what element to generate and consequently to append to already produced ones. However, for series longer than an individual's working memory capacity, at the time people consider only the subsequence of its length. As a result, when they are asked to generate a long random series, they only judge the randomness of its chunks in a rolling window, never evaluating it in the full length. Warren et al. [26] demonstrated that when the capacity limitation is considered the overall frequency of elements in human-generated series compared to the outcomes of the Bernoulli process is similar. However, Biesaga et al. [18] noted that the generation of random series is not a static process and over time the randomness of human-generated series decreases. Furthermore, they demonstrated that this effect is consistent regardless of whether people are asked to produce a long series or an array of short ones. They attributed it to the decay in the attractiveness of the mundane task and linked it to individual differences in cognitive motivation.

Moreover, Kareev [17] famously argued that people would be better off producing random series if only they could either maintain active infinitely long series in their working memory or no series at all. That is because in the former case, they would be able to consider all past choices when judging which element to append. Consequently, the question of whether people were able to generate random series would just be the question of whether they had sufficient knowledge of what the outcomes of a random process look like. On the other hand, if people had no recollection of past choices they would generate every element as an independent. Then, they would not try to simulate the outcomes of a random process but rather the mechanism in which they produced the series would be simply a random process. However, neither of the possibilities is the case in human cognition. As Kareev [25, 42] showed in his later works, there are very good reasons why people only maintain active outcomes of relatively recent events. For example, it allows for faster detection of change in the environment [43–45], 'forgetting' about a partner's defecting strategy in a prison dilemma-like situation [46], and faster recognition of patterns based on a small sample [42]. Therefore, the capacity of working memory might play a role not only during the process of generation of random-like series (limiting the number of active past elements) but also during the learning process as when people observe outcomes of the random process they tend to remember only a limited number of past choices. Therefore, we expect the length of the working memory capacity to be positively correlated with the complexity of human-generated series.

## Hypotheses

Taken together, based on the model proposed by Biesaga et al. [18] in this paper we investigate three main hypotheses:

**Hypothesis 1 (H1)**: *The dynamic of the decay in the performance in random series generation tasks is a function of the visibility of the past choices.*

**Hypothesis 2 (H2)**: *Randomness judgment is positively correlated with the randomness (algorithmic complexity) of human-generated series.*

**Hypothesis 3 (H3)**: *The working memory capacity is positively correlated with the randomness (algorithmic complexity) of human-generated series.*

## Method

Our goal was to test the hypotheses outlined above in two studies. Study 1 was designed to test H1, which predicted that the visibility of past choices would impact the algorithmic complexity of human-generated series, as well as H2, which posited that there is a positive relationship between randomness judgment and the algorithmic complexity of the series. In Study 2, we examined the relationship between the algorithmic complexity of human-generated series with the capacity of working memory (H3).

### Study 1

In the first study, during 15 minutes long procedure participants were asked to produce a binary series of 120 elements and to compare 64 pairs of 7-elements long binary series. They were recruited on *Prolific* and redirected to pavlovia.org where the experiment was run using custom software written in *JavaScript* (it is available on GitHub under MIT License https://gitlab.pavlovia.org/MikoBie/rantool2). The experiment was marked as desktop computers only because it required the usage of the physical keyboard (the *JavaScript* procedure would not launch on mobile devices). The data for the study were collected between June and September of 2021. It was analyzed between January and March 2022.

**Procedure and design.**   The experiment followed a factorial design with one between-subject variable – visibility of the previously generated elements. Participants were assigned to one of the two experimental conditions based on their unique *id* number from *Prolific*. People with odd *id* numbers entered the *visible* condition, in which the last 7 elements of the produced series were visible, while others were assigned to the *invisible* condition in which they did not see any of the previously generated elements. Within each condition, half of the participants first completed the comparison task and afterward the generation task. For the other half of the subjects, the procedure was reversed, they first completed the random series generation task and afterward the comparison task. Regardless of the tasks' order, in the random series production part, the software displayed a red square every 1.25s for .75s. Participants were instructed to imagine a fair coin toss and report the outcome by pressing either the 'comma' or 'dot' key on the keyboard every time they saw the red square (see Fig 2). They were asked in the instruction to use the index and middle fingers of their dominant hand to perform the task.

In the comparison task, participants were asked to compare 64 pairs of 7-elements long series of coin flips. The experiment was run in Polish, where heads is 'orzeł' and tails is 'reszka'. Therefore, the acronyms were O for heads and R for tails. For each couple, they had 5 seconds to decide which series appeared more random. The series differ only by the last element, for example, the displayed series were ORRRROR with ORRRROO. To minimize the effect of the side, we randomly displayed series ending with O either on the right or left side of the screen. Moreover, the order of the displayed series was also assigned at random. Therefore, there were no two participants that saw stimuli in the same order.

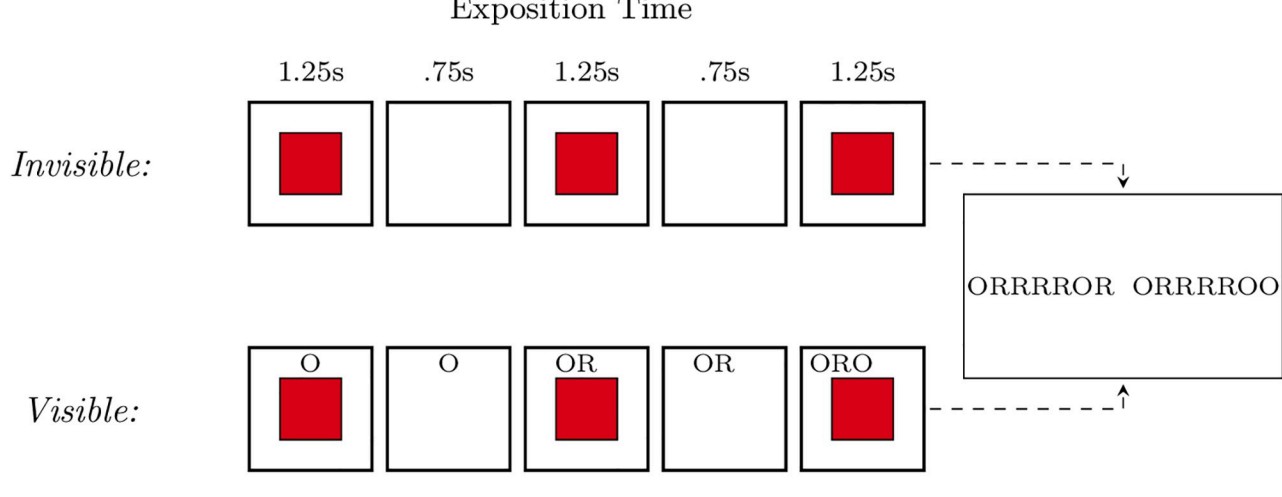

**Fig 2. Flow chart of Study 1 design.** Participants were assigned at random to either invisible or visible experimental conditions. In both conditions, they were presented with a red square every 1.25 s for 0.75 s. In the visible condition, the last seven elements of generated series were displayed on the screen. After the generation phase, in both conditions, participants compared 64 pairs of 7-elements long series of coin flips. For each couple, they had 5 seconds to decide which series is more random. The series differ only by the last digit, for example, the displayed series were ORRRROR with ORRRROO.

**Participants.**    A total number of 199 participants completed the study. They were rewarded with the average rate of £13.55 per hour as payment on *Prolific* for their time. However, after a close examination of the completion times and lengths of the produced series, we decided to exclude some of the records. First, we removed from further analysis records of people who did not follow the task scrupulously and produced significantly shorter series than others. Specifically, observations that were shorter than the typical length of the series—bottom 10% (those with less than 103 elements). Based on this criterion, we removed 17 records. Second, we removed observations with unrealistic times of completion of the entire study (both the random series generation and comparison tasks), that is, 10% of the shortest response time (below 7 minutes and 30 seconds) and responses that extended the assumed maximum completion time (15 minutes). Based on this criterion, we removed additional 31 observations.

Finally, we had 151 (91 males) participants aged from 18 to 61 ($M = 22.4$, $SD = 5.63$). They were assigned to one of the experimental conditions (70 to the *visible* condition) based on their *Prolific id* number. The procedure was approved by the ethics committee of the Robert Zajonc Institute for Social Studies at the University of Warsaw. All participants gave informed consent before taking part in the study. On *Prolific*, before taking part in the study, participants were provided with the short description, aims, and possible risks of the study. They were also informed that at any point in the procedure, they can withdraw their consent to take part in the study or to have their data processed for statistical analysis. Their consent was recorded and stored by *Prolific*. All methods were performed in accordance with relevant guidelines and regulations. As per *Prolific* policy we neither have access to or gathered information that could identify individual participants.

**Data preprocessing.**    Despite removing the records in which participants produced the 10% of the shortest series (shorter than 103 elements) there was still variability in the produced series length. They ranged between 103 and 120 elements (*Median* = 117).

We estimated the algorithmic complexity of participants' series using the 'pybdm' Python module, which implements the Coding Theorem and Block Decomposition methods [47, 48]. The module is available through PyPI at https://pypi.org/project/pybdm/. For statistical analysis, we used *R* programming language [49] with 'mgcv' package for estimating Generalized Additive Mixed Models (GAMMs) [50].

For every participant, we computed two measures of their response's randomness. First, we estimated the overall algorithmic complexity of the entire series. To calculate the overall algorithmic complexity, we averaged the algorithmic complexity of chunks of length 11 because the algorithmic complexity has only been experimentally determined for strings up to 12 elements long [48]. Although the Block Decomposition Method proposed by Zenil et al. [47] can be used to approximate the algorithmic complexity for longer strings, this measure showed less interpersonal variance for our data. That is because the algorithmic complexity calculated with Zenil's method for human-generated data tended to approach the maximum value. Second, for each participant, we calculated a vector of the algorithmic complexity estimates in a moving window of lengths from 5 to 9. This corresponds to the estimated capacity of the working memory that is usually said to be 7 ± 2 elements [51]. In the manuscript we report the results only for the window of length 7 because it matched the length of the sequences in the Comparison Task. The rest can be found in the Supporting Information. Before performing any statistical analysis, we normalized both measures, the overall algorithmic complexity and the rolling algorithmic complexity, using the method described by Zenil et al. [47]. Normalized algorithmic complexity is bounded between 0 and 1, where 0 denotes the simplest possible series, for example, a series that consists of only 0s, and 1 stands for the sequence of a given length that is the most complex.

Additionally, for each participant, we computed the correctness index from the comparison task. It simply represents the proportion of correctly assessed pairs out of all displayed pairs.

**Results and discussion.** In our analysis, as the first step, we tested whether the order affected the results of both tasks. For this purpose, we performed the Wilcoxon Rank Sum test. It allowed for the investigation of whether there were differences in the algorithmic complexity distributions between groups of participants that completed the tasks in different orders. The result revealed that the difference was not significant, $W = 2621$, $p = .483$. Furthermore, we tested whether there was a difference in the correctness index distributions between groups that completed the tasks in different orders. Similarly, we used the Wilcoxon Rank Sum test which result was also not significant, $W = 2901$, $p = .49$. Consequently, we decided to treat both groups as one in further analysis.

As the second step before testing the main hypotheses, we performed the Wilcoxon Rank Sum test to investigate whether there was a difference in the algorithmic complexity distributions between experimental conditions. Specifically, we tested whether the visibility of the last 7 digits affected the normalized overall algorithmic complexity of participants' sequences. The result of the Wilcoxon Rank Sum test was not significant, $W = 2901$, $p = .705$.

In a more detailed analysis that allowed for testing Hypothesis 1, as a dependent variable we had the normalized rolling algorithmic complexity. We performed the Generalized Additive Mixed Model that allowed for the investigation of both linear and non-linear trends in the normalized rolling algorithmic complexity. As the fixed parametric effect, we had the mean difference between experimental conditions with the *invisible* condition as the reference group in dummy coding. For non-parametric effects, we entered a non-linear difference in trends of algorithmic complexity over time between invisible and visible conditions (with the *invisible* condition being the reference) and a non-linear trend of algorithmic complexity over time. Furthermore, we used subject-level random intercepts and slopes for the time trend to model systematic between-subjects differences. We followed Nakagava et al. [52] method to asses the

**Table 1. The generalized additive mixed model specification.**

| Parametric coefficiants | Estimates | Std. Error | t-value | df | p-value |
|---|---|---|---|---|---|
| (Intercept) | .69083 | .01447 | 47.758 | 16279 | < .0001 |
| Linear difference between trend curves (invisible) | .0265440 | .01078348 | 2.46154 | 16279 | .0138 |
| Smooth terms | | edf | Ref.df. | F | p-value |
| Non-parametric trend curve | | 4.165 | 4.165 | 15.561 | <.0001 |
| Non-parametric difference between trend curves (invisible) | | 2.457 | 2.457 | 4.382 | .0282 |

Marginal $R^2$ = 2.08%

Conditional $R^2$ = 42.12%

model's goodness-of-fit. Therefore, we computed the variance retained by fixed effect (marginal $R^2$) and variance retained by the whole model (conditional $R^2$).

The fitted model explained 42.12% of the variance with fixed effects reproducing 2.08% (see Table 1). The non-linear trend of algorithmic complexity over time was significant (cf. Fig 3),

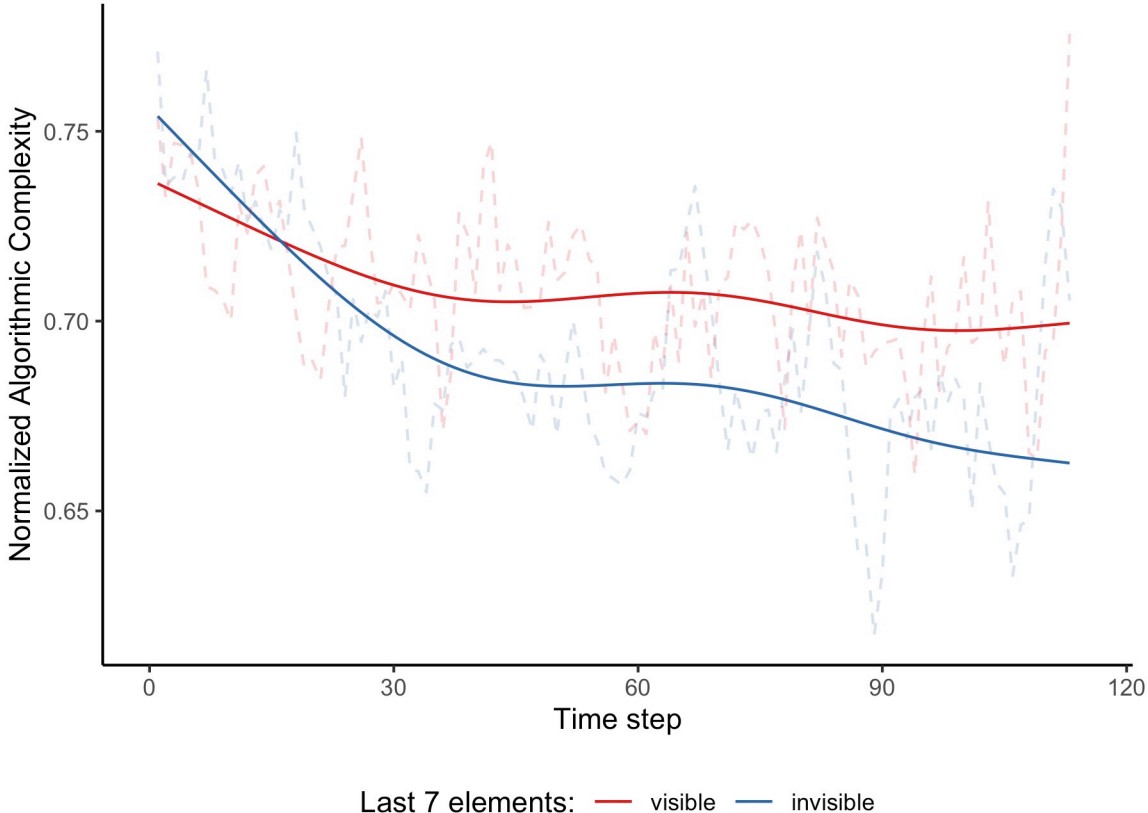

Last 7 elements: — visible — invisible

**Fig 3. Solid lines present trend curves for normalized rolling algorithmic complexity and dashed lines depict the average rolling algorithmic complexity as a function of experimental conditions.** For each participant, we computed vectors of complexity estimates in a rolling window of length 7. Although the experimental task asked for the creation of 120-long series, the length still varied. Therefore, the uncertainty of both the trend curve and the average rolling algorithmic complexity increased around 113th element. Both, the non-linear ($F(edf = 2.457, Ref.df = 2.457) = 4.382$, $p = .028$) and linear difference ($t(16279) = 2.461$, $p = .013$) between the invisible and visible conditions, were significant. The visibility of the last 7 generated elements had the biggest impact at the beginning of the trend curves. In the invisible condition, the effect of the fatigue is much steeper than in the visible condition. Afterward, both curves stabilize at a similar level.

$F(edf = 4.165, Ref.df = 4.165) = 15.561, p < .0001$. Similarly, the non-linear difference between the invisible and visible conditions was significant, $F(edf = 2.457, Ref.df = 2.457) = 4.382$, $p = .028$. Moreover, the linear difference between the conditions was also significant, $t(16279) = 2.461, p = .013$. The visibility of the last 7 generated elements had the biggest impact at the beginning of the trend curves. In the *invisible* condition, the effect of the fatigue is much steeper than in the *visible* condition. Afterward, both curves stabilize at a similar level (compare Fig 3).

In contrast to classic research on the role of the storage component of the working memory random-like series generation [53], our findings did not support the hypothesis that randomness generation is capacity-limited. Similar to the results of Wagenaar [54] and Budescu [13], we did not observe a difference in the level of series randomness between visible and invisible conditions. However, our more detailed analysis revealed a significant difference in the dynamics of the generation process between experimental conditions. Specifically, reducing cognitive load on the working memory storage component led to a slower decrease in performance in the random generation task measured with the algorithmic complexity (see Fig 3). This suggests that participants could maintain a high level of randomness for a longer time when they did not have to solely depend on working memory. This could be due to the fact that both components of working memory rely on the same limited-capacity domain-general central attentional controller [37]. Therefore, reducing the load on the storage component allowed for allocating more cognitive resources to the processing component [37] which is responsible for monitoring past choices and enforcing correction with a new element being appended. However, it did not improve the randomness of the subsequences because whether a person is able to reproduce a random series depends not only on the accuracy of the reproduction but also on the internalized through experience random patterns and consequently randomness judgment mechanism (see Fig 2). This is in line with Schulz et al. [33] results which demonstrated that the patterns found in human-generated data are idiosyncratic.

To test H2, we used a Weighted Least Squares linear regression model (weights were inversely proportional to the variance of the algorithmic complexity at the level of the participant). As a dependent variable, we had the overall (normalized) algorithmic complexity and as the predictor the correctness index. The model was significant, $F(1, 146) = 15.77, p < .001$. The correctness index explained about 9.13% of the algorithmic complexity of the series. With a .1 point increase in the correctness index (the accuracy with which people were able to recognize more complex series in comparison task), there was a .08 (95% *CI* [.04 .12]) increase in the algorithmic complexity of the series (compare Fig 4). This result provides further support for the hypothesis about the positive relationship between the capacity for recognizing more complex series and the ability to generate random-like series. It is consistent with the results of our experimental manipulation and the findings of Falk and Konold [8]. While previous studies [21, 39] have demonstrated the involvement of the central executive system, we emphasize the crucial role of the specific control function responsible for randomness judgment. Furthermore, we posit that if individuals can improve their randomness judgment, they would be able to produce more random series.

Taken together, the results of Study 1 support H1 and H2. The visibility of the last elements of the generated series affected the dynamic of the generation process. Reducing the cognitive load on the working memory storage component allows for the allocation of more attentional resources to the processing component. Consequently, people can maintain the initial level of randomness for a longer time. However, this effect did not have a significant impact on the overall algorithmic complexity of the series. This may be because people's performance in the random generation task is not limited by the allocation of attentional resources to maintaining active past choices but rather by the complexity of representations of the random process they

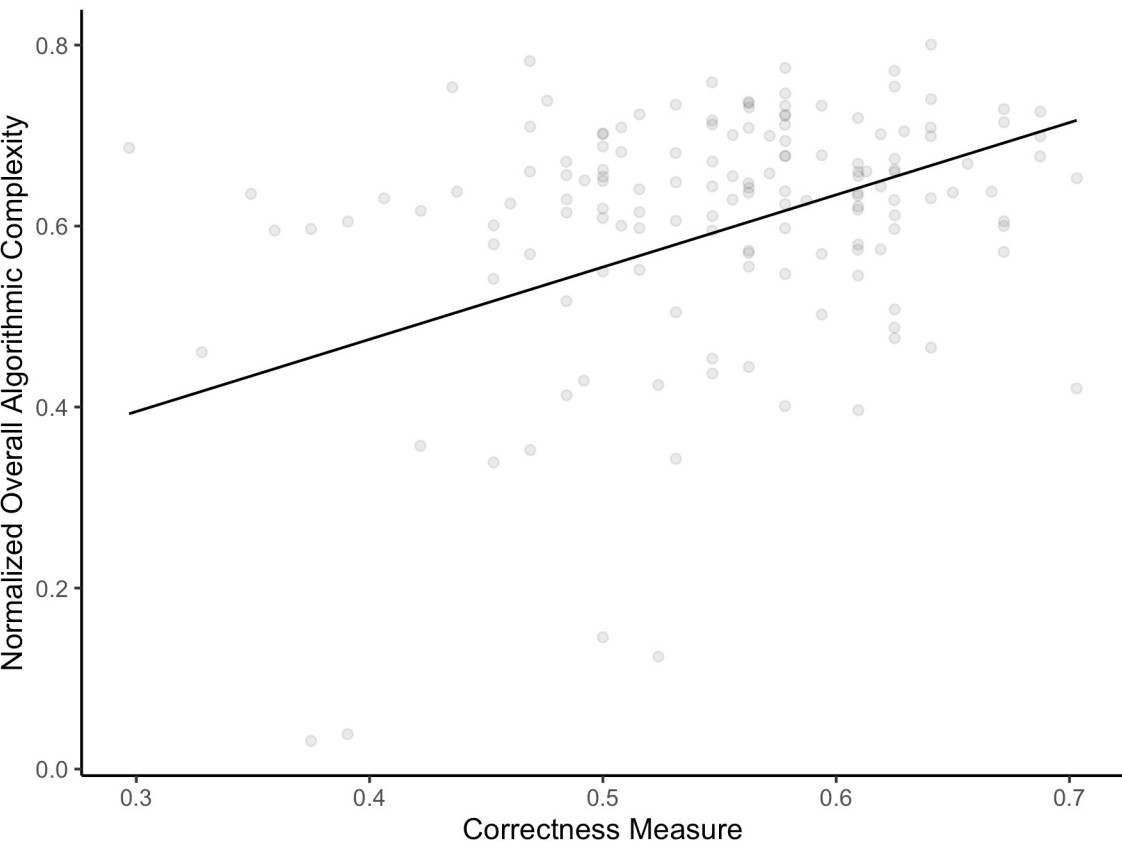

**Fig 4. The trend curve for the relationship between normalized overall algorithmic complexity and the correctness measure, R2 = 9.13%.**

nourish—schemas of the random process they can activate—and the capacity for randomness judgment.

## Study 2

In the second study, during 30 minutes long procedure participants were asked to produce a binary series of 120 elements and to complete the working memory capacity test. Similar to Study 1, they were recruited on *Prolific* (May/June 2022) and redirected to pavlovia.org where the experiment was run using custom software written in *JavaScript* (it is available on GitLab under MIT License https://gitlab.pavlovia.org/MikoBie/ComplexSpan). The experiment was marked as desktop computers only because it required the usage of the physical keyboard (the *JavaScript* procedure would not launch on mobile devices). The data for the study were collected between May and June of 2022. It was analyzed between June and September 2022.

**Procedure and design.**    Although the experiment followed the correlational design we manipulated the order of the tasks. That is because both the random generation task and working memory capacity test are fatigue sensitive [18, 55]. Therefore, half of the participants first completed the random series generation task and later the working memory capacity test while for others the order of the tasks was reversed.

Regardless of the tasks' order, the random series production part followed the same design as in the *visible* condition in Study 1 (see Fig 2). That is, the software displayed a red square

every 1.25 s for .75s. Participants were instructed to imagine a fair coin toss every time they saw a red square and report the outcome by pressing either the 'comma' or 'dot' keys on the keyboard. Similarly to Study 1, they were asked in the instruction to use the index and middle fingers of their dominant hand to perform the task.

For testing working memory capacity, we employed a complex span paradigm [56]. We followed the procedure of the operation span task developed by Turner and Engle [57] in which participants have to decide whether mathematical operations are correct while memorizing unrelated stimuli presented after each operation. Our implementation differ from the classic procedure with one small difference. Instead of using words as the material to memorize we used single consonants. Other than that we followed the classic design. It was based on the *Python* procedure developed by Lau et al. [58].

The main task in the working memory capacity test was to recall a string of letters in the correct order (see right panel of Fig 5 for two letters example). With each correct recall the length of the string to recall in the next trial increased by one letter. In between each displayed letter (each letter was presented for .8s), participants were asked to correctly solve a simple math equation, for example, $(3 \times 8) - 19$. The correct answers in all equations were limited to single digits. The procedure ended with two consequent errors in recall (there was no limit on the length of the string to recall).

The main task—complex recall—was preceded by several trials that were meant to familiarize the participants with every element of the main task. First, participants practiced the simple recall procedure (see the left panel of Fig 5). They were simply asked to recall a string of letters displayed one after another (each letter was displayed for .8s with .5s intervals in-between letters). With two correct recalls the set of letters increased. The procedure ended with three errors on the string of the same length.

Second, participants solved 10 simple equations (the equations used in the trial were not later used in the main task). To complete operations practice, they had to solve at least 65% of the equations correctly. They got feedback after each equation on whether they gave a correct answer (see the middle panel of Fig 5).

Third, participants practiced the complex recall procedure. Their task was to recall two-element long strings (see the right panel of Fig 5). Before each letter was displayed they had to solve a simple equation. The time for giving the correct answer for the simple equation was

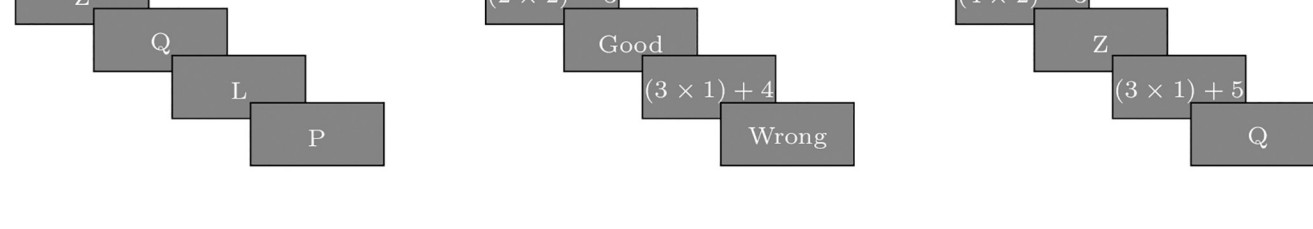

SIMPLE RECALL                OPERATIONS PRACTICE                COMPLEX RECALL

**Fig 5. Flow chart of working memory capacity test.** The left panel depicts the stimuli displayed in the simple recall practice. Each letter was displayed for .8 s with .5 s intervals in-between. The middle panel shows the stimuli displayed in the operations practice. Participants saw 10 equations and after each answer they got feedback whether they solved it correctly. The right panel illustrates the main task. Participants were asked to recall displayed letters in the correct order. Before each letter they had to solve a simple equation. The time for giving the correct answer for the simple equation was limited to the average time of the participant's response in the operations practice plus 2.5 standard deviations.

limited to the average time of the participant's response in the operations practice plus 2.5 standard deviations [59]. If the participant failed to answer during this period the program counted it as a wrong answer and displayed another equation until the correct answer was given.

In the main task that followed the practice procedure, the participants were asked to recall a string of letters in the correct order, and with each correct recall, the length of the string to recall in the next trial increased by one letter. In between each displayed letters, participants were required to correctly solve a simple math equation. The correct answers in all equations were limited to single digits. The procedure ended with two consequent errors in recall. The time for giving the correct answer for the equation was limited in a similar manner as in the practice procedure.

**Participants.**    A total number of 200 participants completed the study. They were rewarded with the average rate of £13.72 per hour as payment on *Prolific* for their time. However, after a close examination of the data we decided to remove some records due to the following reasons:

1. the score in processing task in the complex recall was lower than 85% of correct answers. Based on this criterion we excluded 22 records.

2. the results of the working memory capacity test were suspiciously high or low. Therefore, we excluded 10% of the lowest results (below 4) and 10% of the highest results (above 10). Based on this criteria we excluded further 36 records.

Finally, we had 142 (104 *males*) participants aged from 18 to 53 ($M = 26$, $SD = 7.39$). They were assigned at random to either condition in which the random generation task was preceded ($n = 71$) or followed by the working memory capacity test ($n = 71$). The procedure was approved by the ethics committee of the Robert Zajonc Institute for Social Studies at the University of Warsaw. All participants gave informed consent before taking part in the study. On *Prolific*, before taking part in the study, participants were provided with the short description, aims, and possible risks of the study. They were also informed that at any point in the procedure, they can withdraw their consent to take part in the study or to have their data processed for statistical analysis. Their consent was recorded and stored by *Prolific*. All methods were performed in the accordance with relevant guidelines and regulations. As per *Prolific* policy we neither have access to or gathered information that could identify individual participants.

**Data preprocessing.**    Similarly to Study 1, experimental instruction of the randomness task asked for pressing a relevant key on the exposure of a red square. Therefore, each participant's task was to produce a series of 120 elements. However, some people did not react to all stimuli and pressed the relevant keys less frequently than 120 times. Consequently, the length of the produced sequences ranged from 103 to 120 elements (*Median* = 117).

Herein, like in Study 1, we used 'pybdm' *Python* library (available through PyPI at https://pypi.org/project/pybdm/) to estimate the algorithmic complexity of the participants series. For statistical analysis, we again used *R* programming language [49].

For every participant, we estimated the overall algorithmic complexity of the produced series. Likewise, in Study 1, we calculated it as the average of the normalized algorithmic complexity of chunks of length 11. Therefore, it was bounded between 0 and 1, where 0 indicated the simplest possible series, for example, of all 0s and 1 stood for the most complex one of a given length.

Additionally, for each participant, we calculated the partial span score for complex recall. That is, we calculated the total number of correct recalls. This measure, unlike the absolute span score that gives participants credit only if they did not make mistake in the string of a

given length, is more sensitive and allows for better discrimination between high and low-ability participants [56].

**Results and discussion.** In our analysis, again, as the first step, we tested whether the order affected the results of both tasks. For this purpose, we performed the Wilcoxon Rank Sum test. It allowed for the investigation of whether there was a difference in the algorithmic complexity distributions between groups of participants that completed the tasks in different orders. The result revealed that the difference was not significant, $W = 2603$, $p = .738$. Furthermore, we tested whether there was a difference in the complex recall results' distributions between groups that completed the tasks in different orders. Similarly, we used the Wilcoxon Rank Sum test which result was also not significant, $W = 2193.5$, $p = .173$. As a consequence, we decided to treat both groups as one in further analysis.

In the main analysis, to test H3, we used a Weighted Least Squares linear regression model (weights were inversely proportional to the variance of the algorithmic complexity at the level of participant). As a dependent variable, we had the average (normalized) algorithmic complexity and as the predictor the partial span score for complex recall. The model was significant, $F(1, 140) = 10.47$, $p < .01$. The results in the complex recall task explained about 6.29% of the algorithmic complexity of the series. With a 1-point increase of the partial span score, there was a .021 (95% $CI$ [.008 .034]) increase in the algorithmic complexity of the series (see Fig 6).

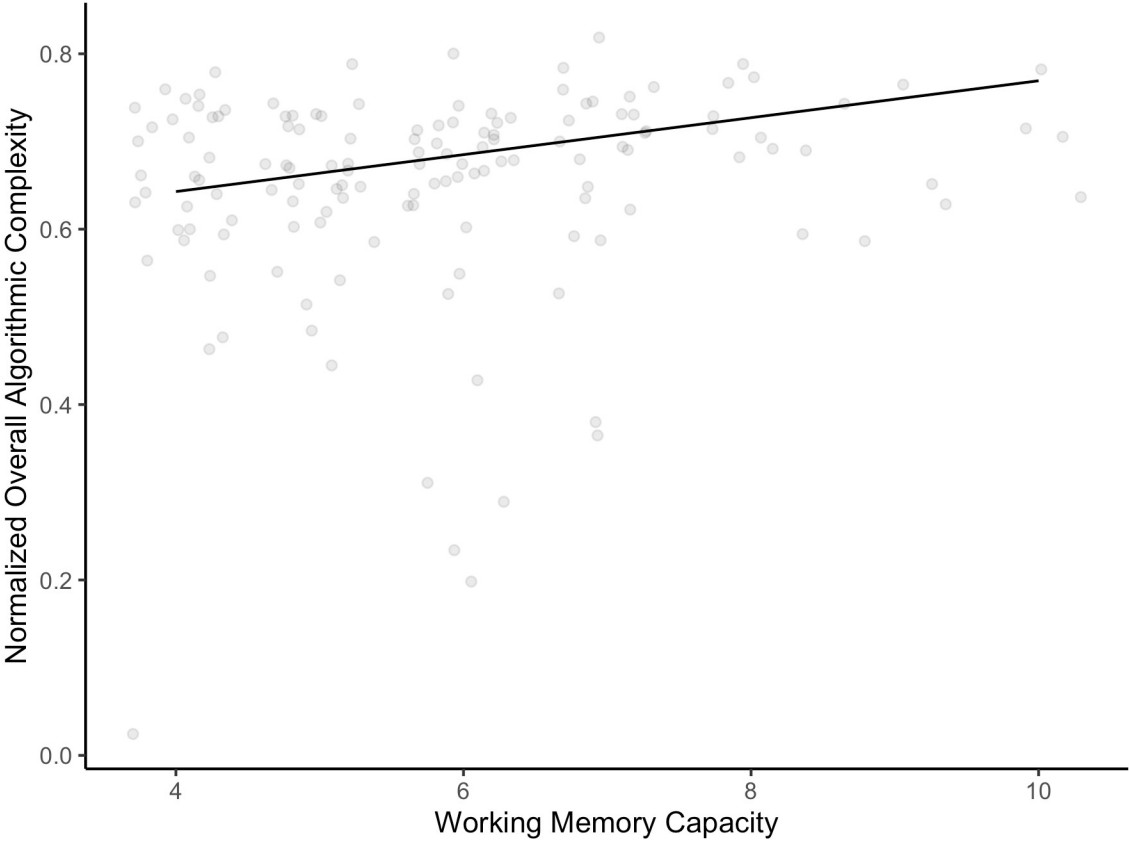

**Fig 6. The trend curve for the relationship between normalized overall algorithmic complexity and the working memory capacity, $R^2 = 6.29\%$.**

This finding supports H3, suggesting that individuals who can maintain longer series in their working memory are capable of generating more complex series. This could be because they are better at examining longer sequences for regularities and avoiding them, as well as inhibiting the appearance of cyclically repeating patterns. Shultz et al. [33] found that certain idiosyncratic patterns tend to systematically reappear in human-generated series, and our results indicate that people with larger working memory capacity are better able to inhibit these patterns for a longer time. Notably, our findings hold even when the cognitive load on working memory is reduced, suggesting that the capacity of working memory is not only crucial for maintaining active the last generated responses but also in the judgment process. This is in line with Rosen and Engle [60] results that demonstrated the role of the working memory capacity in the retrieval process. In general, people with a high working memory capacity are able to retrieve more elements than their compatriots with a low working memory capacity. Therefore, our findings suggest that indeed when producing a random-like series people try to first retrieve and later reproduce memorized patterns. Moreover, individuals with a higher complex span may have better random patterns memorized because they were able to account for more events in the past when observing a random process, like tossing a fair coin. As a result, when they try to reproduce an active schema they are able to produce a series of higher complexity.

## General discussion

The main purpose of this research was to investigate the role of the working memory storage component in the random-like series generation process. We used an objective measure of randomness grounded in AIT—algorithmic complexity. It allowed for tracking the dynamics of the performance in the random series generation task as well as for the investigation of the relationships between the series level of randomness, randomness judgment accuracy, and the capacity of the working memory.

In particular, the results of Study 1, showed that reducing the cognitive load on the working memory storage component did not improve the overall level of randomness of human-generated series but rather prolonged the high-quality performance. In other words, the effect of fatigue reported by Biesaga et al. [18] was smaller when people could have seen their past choices instead of maintaining them active using working memory. We interpret this result in terms of allocating more attentional resources to the processing component of working memory [37]. Consequently, it allowed for maintaining randomness judgment of past choices and accuracy of reproducing active schema of random process on a high level for a longer time. This shows how cognitively expensive during random-like series generation is to constantly maintain active past choices and update them. Therefore, people who chose to do so at the same time are likely to enjoy cognitively demanding tasks [18]. This result also suggests that the limitations of the working memory storage component can not be entirely responsible for people's (in)ability to generate random-like series. That is because even when more attentional resources could have been allocated to the processing component people still were unable to produce significantly more random sequences.

Moreover, the results of Study 1 also suggested that the limitations in human-generated randomness may be attributed to individuals' definition of what constitutes a random sequence. Our results supported the hypothesis that the ability to random-like series generation is tangled to randomness judgment. People who could better distinguish more complex patterns at the same time were generating more random series. We argue that this might be because randomness judgment is embedded in the generation process (see Fig 1 for the schematic model of random series generation). It allows for detecting regularities in produced series and helps to

decide whether the new element should be appended to the past choices, or the active schema shifted and a new item proposed. The judgment is based on personal experience in a specific domain [61]. Therefore, it is not transferable in-between different contexts. This explains both the idiosyncrasy of patterns found in human-generated data [33] and the context dependency of the ability to produce random-like series [18]. People simply have their individual specific to certain domain definition of what randomness looks like and generate series accordingly. Moreover, they use it when retrieving past events that were considered random [23]. It creates a reinforcement loop, in which sequences that they remembered as random have patterns that they consider emblematic of random series. As a result, the limits of people's ability to produce random-like series might be explained in terms of their internalized experience with random outputs. However, it leaves an open question of whether by improving one's understanding of randomness it is possible to improve the ability to produce random-like series in a certain context. Consequently, whether the individual differences in the ability to produce random-like series are constituted by definitions of randomness people nourish.

Schulz et al. [33] demonstrated that the patterns observed in human-generated data are influenced by individual preferences and inhibition processes. Subsequences that individuals attempt to reproduce (or avoid generating) are unique, which allows for identifying a single individual. Our Study 2 results complement those of Study 1 and Schulz et al. [33] by revealing that individual differences in the ability to generate random-like series may also be related to working memory capacity. Participants with higher complex spans were able to produce more random series. We argue that this finding is consistent with the notion that working memory capacity, as proposed in Biesaga et al. [18], reflects the length of the subsequence that people can maintain active and judge for randomness. A longer subsequence allows for detecting more gaps between cyclically repeating patterns in human-generated data, which in turn influences the overall randomness of the series. However, since in Study 2, participants did not have to maintain active in their working memory past choices because the last generated elements were displayed on the screen, we argue that the positive relationship between working memory capacity and the randomness of human-generated data goes beyond the storage component of the working memory. Our results suggest that working memory capacity plays a role during the evaluation of the subsequence and proposing a new element. This might be because people with a higher working memory capacity are able to retrieve longer patterns from the active schema or because the patterns they internalized are of higher complexity.

Our findings can be also interpreted within the broader context of the correlated nature of human behavior. Previous research has demonstrated that repeated human responses tend to follow a specific pattern that is neither highly structured nor completely random. This pattern, known as 1/f scaling or 1/f noise, has been observed across various domains of human behavior, including mental rotation [62], visual search [63], and reaction times during random series generation [64]. According to Kello et al. [27] 1/f scaling is a domain-independent phenomenon that arises from the interaction between different components of the system, resulting in cognitive functions that operate flexibly between ordered and disordered phases.

Therefore, from the 1/f scaling perspective, cognitive functions emerge as metastable patterns of neural and bodily activity, dependent on the optimal level of synchronization and interaction between different components of the system. Biesaga et al. [18] demonstrated that the dynamics of random series production can be explained in terms of the interaction between contextual cues and internal constraints. The optimal systemic interaction between contextual cues, cognitive resources, and motivation to perform well in effortful tasks allows for the production of non-trivial, random-like sequences. However, when fatigue sets in, the system is pushed out of the optimal parameter region, resulting in less random and more patterned series being produced.

In this article, we further examined the cognitive constraints that influence the dynamics of the random-likes series generation process. While reducing the cognitive load on the working memory turned out to allow for maintaining an optimal level of synchronization and interaction between different components of the system for a longer period (see Fig 3) it did not improve the overall randomness of the produced series. Our results suggest that it might be because the definition of the random process people nourish did not change. Therefore, the system was able to function in the optimal state for a longer period but the level of randomness it produces depends not on the constraints of the cognitive system but rather on the internalized patterns of random process outputs. This leaves an open question of whether by improving people's definition of randomness (and providing them with more complex patterns) they will be able to produce more random series. Although some classic studies on random series generation [9, 65] indeed demonstrated that people with a better understanding of such concepts like randomness, probability, or statistics in general performed better in random-likes series generation tasks than novices they only used simple measures based on frequencies of elements and not examined the dynamics of the series complexity.

## Conclusions

Our findings suggest that when individuals rely on external resources to monitor their past actions, they are able to maintain high-quality performance in the random generation task for a longer period. However, reducing the cognitive load on working memory storage does not necessarily lead to an increase in the overall complexity of human-generated data. One possible explanation is that people's personal definitions of randomness, shaped by their domain-specific experiences, often contain flawed and non-random regularities. Thus, individuals with a more refined definition of randomness are better equipped to generate random-like series.

Moreover, we observed that people with a higher working memory capacity were able to consider longer series at a time. This capacity helps them to identify cyclically repeating patterns in human-generated data that are reported by Schulz et al. [66]. As a result, the repeating patterns become more sparse in the series generated by individuals with a higher complex span. However, our results also suggest that the capacity of the working memory influences the judgment of the produced subsequence. This might be due to the fact that people of bigger working memory capacity were able to create better (longer) internalized patterns of random processes which they later activate when producing a random-like series. Therefore, we argue that the limits of people's ability to produce random-like series lie not only in the cognitive constraints but are also compromised by the flawed built-on experience definition of a random process.

## Supporting information

**S1 File.**
(DOCX)

## Acknowledgments

We thank Szymon Talaga for generous theoretical and statistical insights and Magdalena Roszczyńska-Kurasińska for proofreading and general comments. Mikołaj Biesaga would like to express his utmost gratitude to the late Krystyna "Krystylda" Jaczewska for being the best grandmother ever as well as support and patience in the darkest nights of the COVID-19 pandemic.

## Author Contributions

**Conceptualization:** Mikołaj Biesaga, Andrzej Nowak.

**Data curation:** Mikołaj Biesaga.

**Formal analysis:** Mikołaj Biesaga.

**Funding acquisition:** Mikołaj Biesaga.

**Investigation:** Mikołaj Biesaga.

**Methodology:** Mikołaj Biesaga.

**Project administration:** Mikołaj Biesaga.

**Resources:** Mikołaj Biesaga.

**Software:** Mikołaj Biesaga.

**Supervision:** Mikołaj Biesaga, Andrzej Nowak.

**Validation:** Mikołaj Biesaga.

**Visualization:** Mikołaj Biesaga.

**Writing – original draft:** Mikołaj Biesaga.

**Writing – review & editing:** Mikołaj Biesaga.

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
