## [Decision Letter · Decision Letter 0]

18 Sep 2023

PONE-D-23-14835The role of the working memory storage component in a random-like series generationPLOS ONE

Dear Dr. Biesaga,

Thank you for submitting your manuscript to PLOS ONE. After careful consideration, we feel that it has merit but does not fully meet PLOS ONE’s publication criteria as it currently stands. Therefore, we invite you to submit a revised version of the manuscript that addresses the points raised during the review process. Please submit your revised manuscript by Nov 03 2023 11:59PM. If you will need more time than this to complete your revisions, please reply to this message or contact the journal office at plosone@plos.org. Please include the following items when submitting your revised manuscript:A rebuttal letter that responds to each point raised by the academic editor and reviewer(s). You should upload this letter as a separate file labeled 'Response to Reviewers'.A marked-up copy of your manuscript that highlights changes made to the original version. You should upload this as a separate file labeled 'Revised Manuscript with Track Changes'.An unmarked version of your revised paper without tracked changes. You should upload this as a separate file labeled 'Manuscript'.

We look forward to receiving your revised manuscript.

Kind regards,

Iftikhar Ahmed Khan

Academic Editor

PLOS ONE

Journal Requirements:

"We thank Szymon Talaga for generous theoretical and statistical insights and Magda Roszczy´nska-Kurasi´nska for proofreading and general comments. This work was supported by funds from Polish National Science Centre (project no.2019/35/N/HS6/04318) "

"This work was supported by funds from Polish National Science Centre (project no. 2019/35/N/HS6/04318)."

"This work was supported by funds from Polish National Science Centre (project no. 2019/35/N/HS6/04318)."            

Reviewers' comments:

Reviewer's Responses to Questions

**Comments to the Author**

1. Is the manuscript technically sound, and do the data support the conclusions?

Reviewer #1: Yes

Reviewer #2: Partly

2. Has the statistical analysis been performed appropriately and rigorously? 

Reviewer #1: Yes

Reviewer #2: Yes

3. Have the authors made all data underlying the findings in their manuscript fully available?

Reviewer #1: Yes

Reviewer #2: Yes

4. Is the manuscript presented in an intelligible fashion and written in standard English?

Reviewer #1: Yes

Reviewer #2: Yes

5. Review Comments to the Author

Reviewer #1: Working Memory in Randomness Production Review.

This manuscript describes two large randomness production studies. The first study had two parts, one asked for comparative judgments of pairs of binary strings for relative randomness. Participants judgments were correlated with a randomness score. The second study asked participants to produce binary random strings of about 120 charters in length. Participants that saw subsets of their previous choices produced more random strings than those who did not. The second large study asked participants to generate random strings, and correlated the algorithmic complexity of their sequences with their performance on a working memory task. Participants with longer working memory spans did better at producing random series.

I believe this manuscript currently satisfies all 7 criteria for publication in PLOS ONE. Overall, this is a solid, though relatively incremental contribution to the randomness production literature. I say incremental because it is somewhat boxed in by the traditional componential information processing assumptions that undergird the bulk of the randomness production literature. 1.) More emphasis on what participants aren't doing, being random, and less emphasis on what they are doing. 2.) Presumes a Gaussian world. 3.) Somewhat "brainified" and componential, action dynamics aren't considered.

However the project uses a relatively novel dependent measure, algorithmic complexity. the outcomes are interesting because the design successfully manipulates memory load, by allowing subjects to either view or not see their previous responses. Subjects with access to their past responses do a bit better at randomness production.

A recent (May, 2023) JEP:G article on randomness production captured the bulk of participants' production performances by identifying the 62 permutations of 1 to 5 trial histories with Farey ratios. Participants favored shorter, more dynamically stable sequences over longer, less stable sequences. The sequences deviated from classical randomness because they were scaling noises (1/f), that filled the continuum of fractional Gaussian noises, from pink to blue, of which classical "white" randomness comprises only a small slice. Moreover, the empirical patterns were successfully modeled with discrete sine-circle map of the HKB bimanual coordination model.

This manuscript's studies unfold at a slower pace, so coordination dynamics will likely be less prominent, but probably detectable. From a dynamic prospective, the memory manipulation adds a slower timescale feedback loop. Perhaps participants intervene to disrupt perpetually salient patterns? The present version of the manuscript establishes that memory participates in the performances, but it really does not explore how it is used--What are participants using their memories for?

From and action perspective, one ambiguity in the design is that participants pressed two adjacent buttons, so they could either use their right index and right middle fingers, or use their left and right index fingers. From the perspective of oscillatory dynamics this means some for some participants left button presses signified the faster oscillator, and for others it was the right button. In future studies, the response buttons should be separated so there is only one way to place one's index fingers on the response buttons.

I think that conducting counts of all the 62 possible permutations of 1 to 5 trials, and comparing the outcomes with and without working memory would be informative about how cognition is deployed to make the patterns more random. Both conditions can be compared with simulated binary series that are random.

Even if they don't make it into the manuscript, spectral and recurrence analysis could provide information about what participants are doing in each condition, and how they differ. Algorithmic complexity is just too opaque for this purpose.

One minor note, exogenous refers to environmental circumstances, endogenous refers to properties and dynamics intrinsic to the organism. In the manuscript they are equated with top down and bottom up, which would all be endogenous.

Reviewer #2: In this article, the author aims to focus on the dynamic aspect of random-like series production and analyze the role of working memory in the process of series generation. So I have some questions.

1. The structure of this manuscript is confusing. The random generation model part follows the introduction but it seems that this part is a part of the introduction.

2. How do you define/quantify the working memory in your method?

3. In the data preprocessing part, what is processed results?

4. In 306, the moving window is 7, why you set this number?

5. The method of this work is based on hypothesis-testing, so what is the significance?

6. The working memory is related to the age of the participant, however, in your experiment, the age ranges 18 to 55, so in the retrieve process, what is the difference between the young and the elderly?

7. What disciplinary category does this article belong to?

8. you mentioned your finding provide insights into the underlying mechanism of this cognitive process. So what is the insights?

6. PLOS authors have the option to publish the peer review history of their article (what does this mean?). If published, this will include your full peer review and any attached files.

Reviewer #1: No

Reviewer #2: **Yes: **Hongtao Wang

---

## [Author Response · Author response to Decision Letter 0]

24 Nov 2023

Dear Iftikhar Ahmed Khan,

We would like to thank you and the reviewers for the insightful comments and suggestions for the manuscript. We believe that they lead us to necessary improvements in our work. Below, you will find point-by-point replies to the reviewers’ comments. Original comments are in boldface, while our responses are in regular typeface. We added the bibliography for our responses at the end of the file.

Journal Requirements:

We used PLOS ONE’s LaTeX template to meet the requirements.

2. Thank you for stating the following in the Acknowledgments Section of your manuscript: "We thank Szymon Talaga for generous theoretical and statistical insights and Magda Roszczy´nska-Kurasi´nska for proofreading and general comments. This work was supported by funds from Polish National Science Centre (project no.2019/35/N/HS6/04318) " We note that you have provided additional information within the Acknowledgements Section that is not currently declared in your Funding Statement. Please note that funding information should not appear in the Acknowledgments section or other areas of your manuscript. We will only publish funding information present in the Funding Statement section of the online submission form. Please remove any funding-related text from the manuscript and let us know how you would like to update your Funding Statement. Currently, your Funding Statement reads as follows: "This work was supported by funds from Polish National Science Centre (project no. 2019/35/N/HS6/04318)." Please include your amended statements within your cover letter; we will change the online submission form on your behalf.

We removed the statement about the funding from the manuscript and included the amended statement in the cover letter.

3. Thank you for stating the following financial disclosure: "This work was supported by funds from Polish National Science Centre (project no. 2019/35/N/HS6/04318)." Please state what role the funders took in the study. If the funders had no role, please state: "The funders had no role in study design, data collection and analysis, decision to publish, or preparation of the manuscript." If this statement is not correct you must amend it as needed. Please include this amended Role of Funder statement in your cover letter; we will change the online submission form on your behalf. 

We included the amended financial disclosure statement in the cover letter.

Reviewer #1: Working Memory in Randomness Production Review.

This manuscript describes two large randomness production studies. The first study had two parts, one asked for comparative judgments of pairs of binary strings for relative randomness. Participants judgments were correlated with a randomness score. The second study asked participants to produce binary random strings of about 120 charters in length. Participants that saw subsets of their previous choices produced more random strings than those who did not. The second large study asked participants to generate random strings, and correlated the algorithmic complexity of their sequences with their performance on a working memory task. Participants with longer working memory spans did better at producing random series.

I believe this manuscript currently satisfies all 7 criteria for publication in PLOS ONE. Overall, this is a solid, though relatively incremental contribution to the randomness production literature. I say incremental because it is somewhat boxed in by the traditional componential information processing assumptions that undergird the bulk of the randomness production literature. 1.) More emphasis on what participants aren't doing, being random, and less emphasis on what they are doing. 2.) Presumes a Gaussian world. 3.) Somewhat "brainified" and componential, action dynamics aren't considered.

However the project uses a relatively novel dependent measure, algorithmic complexity. the outcomes are interesting because the design successfully manipulates memory load, by allowing subjects to either view or not see their previous responses. Subjects with access to their past responses do a bit better at randomness production.

A recent (May, 2023) JEP:G article on randomness production captured the bulk of participants' production performances by identifying the 62 permutations of 1 to 5 trial histories with Farey ratios. Participants favored shorter, more dynamically stable sequences over longer, less stable sequences. The sequences deviated from classical randomness because they were scaling noises (1/f), that filled the continuum of fractional Gaussian noises, from pink to blue, of which classical "white" randomness comprises only a small slice. Moreover, the empirical patterns were successfully modeled with discrete sine-circle map of the HKB bimanual coordination model.

This manuscript's studies unfold at a slower pace, so coordination dynamics will likely be less prominent, but probably detectable. From a dynamic prospective, the memory manipulation adds a slower timescale feedback loop. Perhaps participants intervene to disrupt perpetually salient patterns? 

Thank you for bringing to our attention Annand and Holden (2022) work. 

Yes, we do believe that this is the case. As we mentioned in the manuscript, Kareev (1992) argued that participants do not consider all possible patterns when producing a random string. Instead, they only take into account the ones that are the most representative of a random series. Therefore, based on their knowledge they avoid certain ‘non-random’ patterns. 

The present version of the manuscript establishes that memory participates in the performances, but it really does not explore how it is used--What are participants using their memories for?

Yes, indeed, in this manuscript, we focus on establishing the involvement of the working memory processing and storage components in the random-like series generation process. However, we mention in the discussion the possible usage of the memories in the task. 

“Our results supported the hypothesis that the ability to random-like series generation is tangled to randomness judgment. People who could better distinguish more complex patterns at the same time were generating more random series. We argue that this might be because randomness judgment is embedded in the generation process (see Fig. 1 for the schematic model of random series generation). It allows for detecting regularities in produced series and helps to decide whether the new element should be appended to the past choices, or the active schema shifted and a new item proposed. The judgment is based on personal experience in a specific domain (Wilke and Barret, 2009). Therefore, it is not transferable in-between different contexts. This explains both the idiosyncrasy of patterns found in human-generated data (Schulz et al., 2021) and the context dependency of the ability to produce random-like series (Biesaga et al., 2021). People simply have their individual specific domain definition of what randomness looks like and generate series accordingly. Moreover, they use it when retrieving past events that were considered random (Olivola and Oppenheimer, 2008). It creates a reinforcement loop, in which sequences that they remember as random have patterns that they consider emblematic of random series. As a result, the limits of people’s ability to produce random-like series might be explained in terms of their internalized experience with random outputs. However, it leaves an open question of whether by improving one’s understanding of randomness it is possible to improve the ability to produce random-like series in a certain context. Consequently, whether the individual differences in the ability to produce random-like series are constituted by definitions of randomness people nourish.”

In future studies, we would like to pursue this path and investigate whether, in an abstract domain, people are able to improve their ability to produce random-like series over training. Some studies suggest that when people receive feedback they are able to outplay computers using random strategy in a zero-sum game (Sharifan, 2016).

From and action perspective, one ambiguity in the design is that participants pressed two adjacent buttons, so they could either use their right index and right middle fingers, or use their left and right index fingers. From the perspective of oscillatory dynamics this means some for some participants left button presses signified the faster oscillator, and for others it was the right button. In future studies, the response buttons should be separated so there is only one way to place one's index fingers on the response buttons.

Thank you for bringing it to our attention. Both studies were performed online and we asked participants in the instruction to use their dominant hand to perform the task. We added this detail to the description of the Procedure and Design of both studies. 

I think that conducting counts of all the 62 possible permutations of 1 to 5 trials, and comparing the outcomes with and without working memory would be informative about how cognition is deployed to make the patterns more random. Both conditions can be compared with simulated binary series that are random.

Thank you for this suggestion. We did perform the analysis. We followed the method described by Annand and Holden (2022), however, none of the average counts in either of the conditions were significantly different from the simulated data. We did not include the analysis in the manuscript, however, we added it in the Supplementary Information. The graph below shows the history counts in two conditions. They are portrayed as differences between mean observed counts and the average count derived from 100 simulated random sequences. 

Figure 1. The empirical visible trial history counts as blue bars and the invisible counts as red bars.

Although the chi-square tests were not significant, we see that in both conditions alternation patterns (10…) seem to be overrepresented while repeating patterns on the other hand are underrepresented (11… or 00…). This is somehow consistent with the results of Annand and Holden (2022). However, there seems to be only one attractor present – antiphase attractor. One explanation might be that the interval between the red square appearance in our study was much longer and the antiphase bimanual attractor destabilizes at around 450 ms (deGuzman and Kelso, 1991). Another explanation could be that our instruction to use the middle and index fingers of the dominant hand could have prevented the emergence of the repetition attractor since fingers of the same hand are too coupled for the appearance of pogo-stick-like button-press resonance in one of them.

Even if they don't make it into the manuscript, spectral and recurrence analysis could provide information about what participants are doing in each condition, and how they differ. Algorithmic complexity is just too opaque for this purpose.

Thank you for this suggestion. Indeed, in the sense of recognizing patterns algorithmic complexity does not allow for understanding what participants are doing during the task. It only gives a coarse measure which allows for concluding on the complexity of the produced series. As we mentioned in the manuscript, the psychological interpretation of this measure might be that one has to allocate more cognitive resources to memorize the more complex series. Therefore, over time, with fatigue, the capacity for the allocation of resources decreases (Biesaga, et al., 2021).

However, we did perform the spectral analysis following Holden's (2005) description. Unsurprisingly, the results were similar to the ones reported by Annand and Holden (2022). In both conditions, we saw a spike in power for alternation patterns. However, the series produced in our study were much shorter than the ones in Annand and Holden's (2022) study, therefore, these results are less reliable. In future studies, we plan on asking participants for the production of more elements so spectral analysis would be more conclusive.

One minor note, exogenous refers to environmental circumstances, endogenous refers to properties and dynamics intrinsic to the organism. In the manuscript they are equated with top down and bottom up, which would all be endogenous.

Thank you, yes, it was our mistake. By exogenous influences, we meant contextual cues while by endogenous cognitive constraints. However, it was misleading to put top-down and bottom-up processes in the brackets because as you rightly pointed out both are endogenous. Therefore, in the corrected manuscript we removed the text in the brackets.

Reviewer #2: In this article, the author aims to focus on the dynamic aspect of random-like series production and analyze the role of working memory in the process of series generation. So I have some questions.

1. The structure of this manuscript is confusing. The random generation model part follows the introduction but it seems that this part is a part of the introduction.

Thank you for the suggestion, we changed the structure according to your remark. The Random Generation Model part is now included in the Introduction while the following subsections are sections.

2. How do you define/quantify the working memory in your method?

We follow the classic Baddeley and Hitch definition of working memory as “(...) a system or systems that are assumed to be necessary in order to keep things in mind while performing complex tasks such as reasoning, comprehension and learning.” (Baddeley, 2010). They proposed a four-component model that included an attentional controller and three modality-based temporary stores (phonological loop, visual-spatial sketch pad, and episodic buffer). All four elements are assumed to interact (Baddeley, Hitch, and Allen, 2009). Based on this model, the works of Biesaga et al. (2021) and Vandierendonck et al. (2012) who argued that the storage component of the working memory must play a distinct role from the central executive system, we designed our method for the measurement of the role of working memory storage and processing components. 

Based on the model proposed by Biesaga et al. (2021), we assumed that the storing component is responsible for maintaining active the subsequence of the already generated sequence, while the processing component is employed to evaluate whether the proposed new element increases the randomness of past choices (compare Fig 1 in the manuscript). Accordingly, we designed Study 1, in which, we manipulated the visibility of the past elements in the random-like series generation task. This manipulation was aimed to decrease the cognitive load on the working memory. On the other hand, the role of the processing component of the working memory was tested in the Comparison Task, in which participants were asked to compare two sequences that only differed with the last elements.

For testing working memory capacity, we employed a complex span paradigm (Conway, 2005). We followed the procedure of the operation span task developed by Turner and Engle (1989) in which participants have to decide whether mathematical operations are correct while memorizing unrelated stimuli presented after each operation. Our implementation differs from the classic procedure with one small detail. Instead of using words as the material to memorize we used single consonants. Other than that we followed the classic design. It was based on the Python procedure developed by Lau et al. (2019).

3. In the data preprocessing part, what is processed results?

For both studies, we described in detail the preprocessing procedure of raw data in the Data preprocessing subsections of the manuscript. The reproducible code and raw data are available on Open Science Framework at http://doi.org/10.17605/OSF.IO/CK78N. In a nutshell, in Study 1, after the processing of the raw results we had three measures: the overall algorithmic complexity measure, a vector of algorithmic complexity estimates in a moving window of length 7, and the correctness index. The first two measures were derived from the random generation task (a sequence of 0s and 1s) while the correctness index was the ratio of correct answers to all displayed pairs in the Comparison Task.

In Study 2, after the processing of the raw results we had two measures: the overall algorithmic complexity and partial span score of complex recall. The former was derived from the random generation task (a sequence of 0s and 1s) while the latter was the number of correct recalls in the complex recall task.

4. In 306, the moving window is 7, why you set this number?

In the manuscript, we added the motivation behind setting the window size to 7. The length of the moving window was meant to correspond to the length of the sequence in the comparison task (where participants decided which of the 7-elements long sentences seemed to be more random). We decided on 7-element long sequences because the estimated capacity of the working memory is usually said to be 7 ± 2 elements (Baddeley, 1986). Moreover, in the manuscript, we present the results only for the window of length 7 while in the Supporting Information, we also compute results for windows length between 5, 6, 8, and 9.

5. The method of this work is based on hypothesis-testing, so what is the significance?

In terms of the significance level, for each test performed, we report its significance level in the manuscript. As per tradition, in Psychology, the probability below .05 allows rejecting the null hypothesis of no effect.

6. The working memory is related to the age of the participant, however, in your experiment, the age ranges 18 to 55, so in the retrieve process, what is the difference between the young and the elderly?

Thank you for this question. It helped us to realize that there was a typo in the upper range of age (it should have been 53 in the second study). However, in a great scheme of things, this detail does not change our response to your question. We did not include age in the model because we just had 30 responses from participants older than 30 years (8 over the age of 40). Therefore, it would be very difficult to draw conclusions from such a sample because the decline in working memory capacity begins in the mid-20s (i.e., Park et al., 2002) and we did not have enough data points for older participants to control the variance. Moreover, the ability to produce random-like series also decreases over age. Gauvrit et al. (2017), in an online study of 3400 participants, showed that the developmental curve of the ability to produce random-like series is similar to most cognitive abilities. It peaks at the age of 25 and decreases slowly afterward and more quickly after the age of 60.

7. What disciplinary category does this article belong to?

We believe it belongs to Cognitive Psychology and it contributes to the literature on randomness production. 

8. you mentioned your finding provide insights into the underlying mechanism of this cognitive process. So what is the insights?

We summarize the insight in the Conclusions subsection of the article. 

“Our findings suggest that when individuals rely on external resources to monitor their past actions, they are able to maintain high-quality performance in the random generation task for a longer period. However, reducing the cognitive load on working memory storage does not necessarily lead to an increase in the overall complexity of human-generated data. One possible explanation is that people's personal definitions of randomness, shaped by their domain-specific experiences, often contain flawed and non-random regularities. Thus, individuals with a more refined definition of randomness are better equipped to generate random-like series.

Moreover, we observed that people with a higher working memory capacity were able to consider longer series at a time. This capacity helps them to identify cyclically repeating patterns in human-generated data that are reported by Schulz et al. (2012). As a result, the repeating patterns become more sparse in the series generated by individuals with a higher complex span. However, our results also suggest that the capacity of the working memory influences the judgment of the produced subsequence. This might be due to the fact that people of bigger working memory capacity were able to create better (longer) internalized patterns of random processes which they later activate when producing a random-like series. Therefore, we argue that the limits of people's ability to produce random-like series lie not only in the cognitive constraints but are also compromised by the flawed built-on experience definition of a random process.”

Bibliography

Annand, C. T., & Holden, J. G. (2022). Embodied nonlinear dynamics of cognitive performance. Journal of Experimental Psychology: General. https://doi.org/10.1037/xge0001319

Baddeley, A. (2010). Working memory. Current Biology, 20(4), R136–R140. https://doi.org/10.1016/j.cub.2009.12.014

Baddeley, A. D. (1986). Working memory. England: Oxford University Press.

Baddeley, A.D., Hitch, G.J., and Allen, R.J. (2009). Working memory and binding in sentence recall. Journal of memory and Language. 61, 438–456.

Biesaga, M., Talaga, S., & Nowak, A. (2021). The Effect of Context and Individual Differences in Human-Generated Randomness. Cognitive Science, 45(12), e13072. https://doi.org/10.1111/cogs.13072

Conway, A. R. A., Kane, M. J., Bunting, M. F., Hambrick, D. Z., Wilhelm, O., & Engle, R. W. (2005). Working memory span tasks: A methodological review and user’s guide. Psychonomic Bulletin & Review, 12(5), 769–786. https://doi.org/10.3758/BF03196772

deGuzman, G. C., & Kelso, J. A. S. (1991). Multifrequency behavioral patterns and the phase attractive circle map. Biological Cybernetics, 64(6), 485–495. https://doi.org/10.1007/BF00202613

Gauvrit, N., Zenil, H., Soler-Toscano, F., Delahaye, J.-P., & Brugger, P. (2017). Human behavioral complexity peaks at age 25. PLOS Computational Biology, 13(4), e1005408. https://doi.org/10.1371/journal.pcbi.1005408

Holden, J. G. (2005). Gauging the fractal dimension of cognitive performance. In M. A. Riley & G. C. Van Orden (Eds.), Tutorials in contemporary nonlinear methods for the behavioral sciences (pp. 267-319). Retrieved from http://www.nsf.gov/sbe/bcs/pac/nmbs/nmbs.jsp

Kareev, Y. (1992). Not that bad after all: Generation of random sequences. Journal of Experimental Psychology: Human Perception and Performance, 18(4), 1189–1194. https://doi.org/10.1037/0096-1523.18.4.1189

Lau, Z. J., Pham, T. T., & Makowski, D. (2019). neuropsychology/ComplexSpan: 0.0.1 [Computer software]. Zenodo. https://doi.org/10.5281/zenodo.3529329

Olivola, C. Y., & Oppenheimer, D. M. (2008). Randomness in retrospect: Exploring the interactions between memory and randomness cognition. Psychonomic Bulletin & Review, 15(5), 991–996. https://doi.org/10.3758/PBR.15.5.991

Park, D. C., Lautenschlager, G., Hedden, T., Davidson, N. S., Smith, A. D., & Smith, P. K. (2002). Models of visuospatial and verbal memory across the adult life span. Psychology and Aging, 17(2), 299–320. https://doi.org/10.1037/0882-7974.17.2.299

Schulz, M.-A., Baier, S., Timmermann, B., Bzdok, D., & Witt, K. (2021). A cognitive fingerprint in human random number generation. Scientific Reports, 11(1), Article 1. https://doi.org/10.1038/s41598-021-98315-y

Schulz, M.-A., Schmalbach, B., Brugger, P., & Witt, K. (2012). Analysing Humanly Generated Random Number Sequences: A Pattern-Based Approach. PLOS ONE, 7(7), e41531. https://doi.org/10.1371/journal.pone.0041531

Sharifian, S. (2016). Random Number Generation using Human Gameplay [Master thesis, Graduate Studies]. https://doi.org/10.11575/PRISM/27524

Turner, M. L., & Engle, R. W. (1989). Is working memory capacity task dependent? Journal of Memory and Language, 28(2), 127–154. https://doi.org/10.1016/0749-596X(89)90040-5

Vandierendonck, A., Demanet, J., Liefooghe, B., & Verbruggen, F. (2012). A chain-retrieval model for voluntary task switching. Cognitive Psychology, 65(2), 241–283. https://doi.org/10.1016/j.cogpsych.2012.04.003

Wilke, A., & Barrett, H. C. (2009). The hot hand phenomenon as a cognitive adaptation to clumped resources. Evolution and Human Behavior, 30(3), 161–169. https://doi.org/10.1016/j.evolhumbehav.2008.11.004

---

## [Decision Letter · Decision Letter 1]

13 Dec 2023

PONE-D-23-14835R1The role of the working memory storage component in a random-like series generationPLOS ONE

Dear Dr. Biesaga,

Thank you for submitting your manuscript to PLOS ONE. After careful consideration, we feel that it has merit but does not fully meet PLOS ONE’s publication criteria as it currently stands. Therefore, we invite you to submit a revised version of the manuscript that addresses the points raised during the review process.

We look forward to receiving your revised manuscript.

Kind regards,

Iftikhar Ahmed Khan

Academic Editor

PLOS ONE

Journal Requirements:

Reviewers' comments:

Reviewer's Responses to Questions

**Comments to the Author**

1. If the authors have adequately addressed your comments raised in a previous round of review and you feel that this manuscript is now acceptable for publication, you may indicate that here to bypass the “Comments to the Author” section, enter your conflict of interest statement in the “Confidential to Editor” section, and submit your "Accept" recommendation.

Reviewer #1: All comments have been addressed

Reviewer #2: (No Response)

2. Is the manuscript technically sound, and do the data support the conclusions?

Reviewer #1: Yes

Reviewer #2: (No Response)

3. Has the statistical analysis been performed appropriately and rigorously? 

Reviewer #1: Yes

Reviewer #2: (No Response)

4. Have the authors made all data underlying the findings in their manuscript fully available?

Reviewer #1: Yes

Reviewer #2: (No Response)

5. Is the manuscript presented in an intelligible fashion and written in standard English?

Reviewer #1: Yes

Reviewer #2: (No Response)

6. Review Comments to the Author

Reviewer #1: I'm satisfied that the revisions address the issues and questions raised in the first round of reviews. I recommend accepting this revised version of the manuscript for publication.

Reviewer #2: I have two more questions.

1. in page 10/22, line 384-385, the author mentioned that Specifically, reducing cognitive load on the working memory storage 384 component led to a slower decrease in performance (see Fig. 3). However in fig3, it is the relationship between the algorithmic complexity and the time step not the performance and the time, what is the reason?

2. the generation of random series in the study is based on the binary series, as we know, the working memory can be tested with other protocols for example the n-back experiment. so why the author choose this experimental protocol?

3. There are some works also related to working memory and the analysis of working memory is based on neural activity within the brain. The author can have a look at these.

T. Xu, J. Huang, Z. Pei, J. Chen, J. Li, A. Bezerianos, N. Thakor, and H. Wang, "The Effect of Multiple Factors on Working Memory Capacities: Aging, Task Difficulty, and Training," IEEE Transactions on Biomedical Engineering, vol. 70, no. 6, pp. 1967-1978, 2023, doi: 10.1109/TBME.2022.3232849.

Z. Pei, H. Wang, A. Bezerianos, and J. Li, "EEG-Based Multi-Class Workload Identification Using Feature Fusion and Selection," IEEE Transactions on Instrumentation and Measurement, vol. PP, pp. 1-1, 08/27 2020, doi: 10.1109/TIM.2020.3019849.

Z. Pei, T. Xu, A. Bezerianos, J. Li, Y. Sun, and H. Wang, The Effect of Longitudinal Training on Working Memory Capacities: An Exploratory EEG Study. 2020.

7. PLOS authors have the option to publish the peer review history of their article (what does this mean?). If published, this will include your full peer review and any attached files.

Reviewer #1: No

Reviewer #2: **Yes: **Hongtao Wang

---

## [Author Response · Author response to Decision Letter 1]

15 Dec 2023

Dear Iftikhar Ahmed Khan,

We would like to thank you and the reviewers for the insightful comments and suggestions for the manuscript. We believe that they lead us to necessary improvements in our work. Below, you will find point-by-point replies to the reviewers’ comments. Original comments are in boldface, while our responses are in regular typeface. 

Journal Requirements:

We revised the reference list and it is complete and correct.

Reviewer #1: I'm satisfied that the revisions address the issues and questions raised in the first round of reviews. I recommend accepting this revised version of the manuscript for publication.

Thank you.

Reviewer #2: I have two more questions.

1. in page 10/22, line 384-385, the author mentioned that Specifically, reducing cognitive load on the working memory storage 384 component led to a slower decrease in performance (see Fig. 3). However in fig3, it is the relationship between the algorithmic complexity and the time step not the performance and the time, what is the reason?

Thank you for pointing this out. Indeed, the abovementioned sentence could have been misleading. By “performance” we meant performance in the random-like series generation task which we measured with algorithmic complexity. We added this detail in the corrected version of the manuscript.

2. the generation of random series in the study is based on the binary series, as we know, the working memory can be tested with other protocols for example the n-back experiment. so why the author choose this experimental protocol?

We decided to use a task based on the complex span tasks procedure developed by Turner and Engle (1989) for the measurement of working memory capacity because the structure of the task was similar to the Biesaga et al. (2021) random-like series generation model. Complex span tasks demand serial recall, whereby participants retrieve items using only self-generated cues (Kane et al., 2007). Biesaga et al.'s (2021) model also assumes sequential updating of elements highlighted in the working memory storage component without external cues. On the other hand, for example, n-back tasks require recognition, whereby participants discriminate target items from familiar foils (Kane et al., 2007). Therefore, we found a measure based on complex span tasks as more ecologically valid for testing the relationship between working memory capacity and the ability to generate random-like series.

However, certainly, in future research, it would be of utmost interest to further explore this relationship using different working memory capacity measures. Especially, as the scientific literature provides the evidence that for example complex span tasks and n-back tasks might measure different aspects of the working memory capacity (i.e., Schmiedek et al., 2014; Scharinger et al., 2017).

Biesaga, M., Talaga, S., & Nowak, A. (2021). The Effect of Context and Individual Differences in Human-Generated Randomness. Cognitive Science, 45(12), e13072. https://doi.org/10.1111/cogs.13072

Kane, M., Conway, A., Miura, T., & Colflesh, G. (2007). Working Memory, Attention Control, and the N-Back Task: A Question of Construct Validity. Journal of Experimental Psychology. Learning, Memory, and Cognition, 33, 615–622. https://doi.org/10.1037/0278-7393.33.3.615

Scharinger, C., Soutschek, A., Schubert, T., & Gerjets, P. (2017). Comparison of the Working Memory Load in N-Back and Working Memory Span Tasks by Means of EEG Frequency Band Power and P300 Amplitude. Frontiers in Human Neuroscience, 11. https://www.frontiersin.org/articles/10.3389/fnhum.2017.00006

Schmiedek, F., Lövdén, M., & Lindenberger, U. (2014). A task is a task is a task: Putting complex span, n-back, and other working memory indicators in psychometric context. Frontiers in Psychology, 5. https://www.frontiersin.org/articles/10.3389/fpsyg.2014.01475

Turner, M. L., & Engle, R. W. (1989). Is working memory capacity task dependent? Journal of Memory and Language, 28(2), 127–154. https://doi.org/10.1016/0749-596X(89)90040-5

3. There are some works also related to working memory and the analysis of working memory is based on neural activity within the brain. The author can have a look at these.

T. Xu, J. Huang, Z. Pei, J. Chen, J. Li, A. Bezerianos, N. Thakor, and H. Wang, "The Effect of Multiple Factors on Working Memory Capacities: Aging, Task Difficulty, and Training," IEEE Transactions on Biomedical Engineering, vol. 70, no. 6, pp. 1967-1978, 2023, doi: 10.1109/TBME.2022.3232849.

Z. Pei, H. Wang, A. Bezerianos, and J. Li, "EEG-Based Multi-Class Workload Identification Using Feature Fusion and Selection," IEEE Transactions on Instrumentation and Measurement, vol. PP, pp. 1-1, 08/27 2020, doi: 10.1109/TIM.2020.3019849.

Z. Pei, T. Xu, A. Bezerianos, J. Li, Y. Sun, and H. Wang, The Effect of Longitudinal Training on Working Memory Capacities: An Exploratory EEG Study. 2020.

Thank you for bringing these publications to our attention. In future studies, we plan on using EEG to further investigate the usage of working memory in the process of random-like series generation.

---

## [Editor Report · Decision Letter 2]

19 Dec 2023

The role of the working memory storage component in a random-like series generation

PONE-D-23-14835R2

Dear Dr. Biesaga,

We’re pleased to inform you that your manuscript has been judged scientifically suitable for publication and will be formally accepted for publication once it meets all outstanding technical requirements.

Kind regards,

Iftikhar Ahmed Khan

Academic Editor

PLOS ONE
---

## [Editor Report · Acceptance letter]

9 Jan 2024

PONE-D-23-14835R2 

PLOS ONE

Dear Dr. Biesaga, 

I'm pleased to inform you that your manuscript has been deemed suitable for publication in PLOS ONE. Congratulations! Your manuscript is now being handed over to our production team.

Kind regards, 

on behalf of

Dr. Iftikhar Ahmed Khan 

Academic Editor

PLOS ONE